

# NoahPy: A differentiable Noah land surface model for simulating permafrost thermo-hydrology

Wenbiao Tian[1,2], Hu Yu[3], Shuping Zhao[2*], Yuhe Cao[4], Wenjun Yi[2], Jiwei Xu[2], Zhuotong Nan[1,2,5*]

[1]State Key Laboratory of Climate System Prediction and Risk Management, Nanjing Normal University, Nanjing 210023, China
[2]Key Laboratory of Ministry of Education on Virtual Geographic Environment, Nanjing Normal University, Nanjing 210023, China
[3]North Information Control Group, Nanjing 211161, China
[4]College of Forestry, Northeast Forestry University, Harbin 150040, China
[5]Jiangsu Center for Collaborative Innovation in Geographical Information Resource Development and Application, Nanjing, 210023, China

*Correspondence to*: shuping@njnu.edu.cn; nanzt@njnu.edu.cn

**Abstract.** Accurately representing permafrost in Earth System Models is a grand challenge that creates major uncertainty. A promising path forward is to create hybrid models that synergize process-based physics with deep learning, but this is fundamentally hindered by the non-differentiable nature of traditional land surface models (LSMs), which are incompatible with modern AI workflows. To overcome this limitation, we present NoahPy, a fully differentiable LSM developed by reconstructing the Noah LSM's governing partial differential equations into a process-encapsulated Recurrent Neural Network (RNN). We first demonstrate that NoahPy perfectly replicates the numerical behaviour of the modified Noah LSM, achieving Nash-Sutcliffe Efficiency (NSE) coefficients above 0.99 for both soil temperature and liquid water. We then show that at a permafrost site, the calibrated NoahPy achieves robust simulation performance for for soil temperature (NSE > 0.9) and liquid water (NSE > 0.8). Critically, the differentiable workflow, when combined with the Adam optimizer, is significantly faster, more stable, and yields simulations with lower uncertainty compared to traditional SCE-UA calibration algorithm. NoahPy thus provides a foundational, "glass-box" framework that closes a key technical gap, enabling the development of the next generation of hybrid AI-physics models needed to more reliably predict the future of the cryosphere.

## 1 Introduction

The advent of deep learning has catalyzed a paradigm shift in Earth system science. Large-scale, data-driven models like Google DeepMind's GraphCast (Lam et al., 2023) and Huawei's Pangu-Weather (Bi et al., 2023) demonstrate remarkable skill in Earth system forecasting. However, their predictive power is often shadowed by a critical limitation: as "black-box" systems, they offer no guarantee of physical consistency or interpretability (Nearing et al., 2021; Wi and Steinschneider, 2022). While techniques from eXplainable AI (XAI) can provide post-hoc insights (Rudin, 2019; O'loughlin et al., 2025), they cannot enforce physical laws, creating the risk of learning statistically powerful but mechanistically flawed



relationships. This "physics gap" has spurred a movement towards hybrid modeling that synergize the predictive prowess of machine learning with the mechanistic rigor of process-based physical models (Irrgang et al., 2021; Reichstein et al., 2019; Chen et al., 2023).

A powerful approach in this domain is the physics-informed neural network, which embeds the governing equations of a physical system directly into the model's architecture (Reichstein et al., 2019; Chen et al., 2023). Unlike "loosely-coupled" hybrids that use physics as a soft penalty in the loss function (Wang et al., 2020; Xie et al., 2022) or use machine learning to correct a physical model's output (Bonavita and Laloyaux, 2020), this deeply-integrated approach imposes hard constraints, rendering the model structurally incapable of violating fundamental laws. The primary obstacle to this integration has been

technical: most established geophysical models are implemented as non-differentiable numerical solvers, making them incompatible with the gradient-based optimization central to deep learning (Rumelhart et al., 1986). A transformative solution is differentiable programming, which involves rewriting a physical model's logic using differentiable operations within a machine learning framework like PyTorch or TensorFlow. This recasts the physical model into a "glass-box" system that is both physically interpretable and trainable end-to-end via backpropagation (Shen et al., 2023). Recent successes in

hydrology have demonstrated the potential of this approach, yielding models with higher accuracy and improved generalization (Feng et al., 2022; Wang et al., 2024).

     This approach is particularly critical for modeling permafrost. Improving the representation of these processes in Earth system models is a grand challenge (Schädel et al., 2024). Covering nearly 15% of the Northern Hemisphere's exposed land area, permafrost is a crucial regulator of global water, energy, and carbon cycles (Obu, 2021). Despite its vast scale, state-of-

the-art land surface models (LSMs), as the foundational components of climate models, have well-documented deficiencies in representing freeze-thaw processes in these regions (Matthes et al., 2025; Abdelhamed et al., 2023). They often simplify or omit key thermo-hydrological dynamics, such as abrupt thaw (thermokarst), the formation of excess ground ice, the insulation from thick organic soil layers, and complex water transport at the freeze-thaw front (cryosuction). These simplifications lead to significant biases in simulating active layer dynamics and the rate of permafrost thaw, and low

confidence in the timing and magnitude of the permafrost carbon feedback, undermining the reliability of climate projections and estimates of the remaining carbon budget.

     A differentiable LSM, by itself, does not inherently fix these physical deficiencies. Its true power is unlocked when applied to an already improved physical core, enabling it to serve as a foundational component for more sophisticated hybrid artificial intelligence (AI) systems. A differentiable, permafrost-focused LSM enables AI-driven parameterization, where the

differentiable LSM is coupled with a neural network that learns to predict its internal parameters (e.g., hydraulic conductivity, thermal properties) from external data, thus addressing the long-standing challenge of parameter uncertainty (Tsai et al., 2021; Wang et al., 2024; Sun et al., 2024). More importantly, it can be embedded as a physics core within a larger, end-to-end trainable AI-based Earth system model. This forces the larger model to follow the laws of land surface physics, providing essential bounds for its predictions in data-scarce permafrost regions.





Therefore, creating a differentiable permafrost-focused LSM is not an incremental step but a necessary foundation for the next generation of hybrid Earth system models. To address this gap, we introduce NoahPy: a fully differentiable land surface model specifically improved for simulating permafrost thermo-hydrology. We have rewritten the widely-used, Fortran-based Noah LSM into a differentiable Python framework by encapsulating its governing partial differential equations within a Recurrent Neural Network (RNN) structure. This novel implementation preserves the complete

mechanistic integrity of the original model while unlocking the full power of gradient-based optimization.

## 2 Material and methods

### 2.1 The modified Noah LSM

The Noah LSM (v3.4.1) (Chen et al., 1997) is a widely used model that simulates one-dimensional thermo-hydrological transport within the atmosphere-vegetation-soil continuum. It serves as the land-surface module in prominent

systems like the Weather Research and Forecasting (WRF) model (Ek et al., 2003) and the Global Land Data Assimilation System (GLDAS) (Rodell et al., 2004). In the Noah LSM, the governing equation for soil heat transfer is the one-dimensional heat conduction equation:

$$C_s \frac{\partial T_s}{\partial t} = \frac{\partial}{\partial z}\left(\lambda \frac{\partial T_s}{\partial z}\right) + Q \tag{1}$$

where $T_s$ is the soil temperature (K), $t$ is time (s), $z$ is soil depth (m), $C_s$ is the volumetric soil heat capacity (J·m$^{-3}$·K$^{-1}$), $\lambda$ is

the soil thermal conductivity (W·m$^{-1}$·K$^{-1}$), and $Q$ represents the source/sink term (W·m$^{-3}$), such as the latent heat of fusion during ice-water phase change. The soil heat capacity, $C_s$ , is calculated as a weighted sum of its constituents:

$$C_s = \theta C_w + (1 - \theta_s) C_{soil} + (\theta_s - \theta) C_{air} \tag{2}$$

where $\theta$ is the volumetric liquid water content (m$^3$·m$^{-3}$), $\theta_s$ is the saturated volumetric water content (m$^3$·m$^{-3}$), and $C_w$, $C_{soil}$, and $C_{air}$ are the heat capacities of water, soil solids, and air, respectively.

Liquid water movement in the soil is simulated by the Richards' equation (Chen et al., 1996):

$$\frac{\partial \theta}{\partial t} = \frac{\partial}{\partial z}\left[D(\theta) \frac{\partial \theta}{\partial z}\right] + \frac{\partial K(\theta)}{\partial z} + S(\theta) \tag{3}$$

where $D = K(\theta) \frac{\partial \Psi}{\partial \theta}$ , known as the soil-water diffusivity (m$^2$·s$^{-1}$), $K$ is the hydraulic conductivity (m·s$^{-1}$), $\Psi$ is the soil matric potential (m). $S$ represents water sources and sinks (s$^{-1}$) (e.g., infiltration and evapotranspiration). The empirical soil hydraulic scheme proposed by Campbell (1974) is utilized to parameterize $\Psi$–$\theta$ and K–$\theta$, relationships :

$$K(\theta) = K_s \left(\frac{\theta}{\theta_s}\right)^{2b+3} \tag{4}$$



$$\Psi(\theta) = \Psi_s \left( \frac{\theta}{\theta_s} \right)^{-b} \tag{5}$$

where $K_s$ represent the saturated hydraulic conductivity (m·s$^{-1}$), $\Psi_s$ is the soil water potential at air entry (m), and b is an empirical parameter (dimensionless) related to the pore size distribution of the soil matrix.

For this study, we used a version of the Noah LSM specifically modified for permafrost applications (Chen et al., 2015; Wu et al., 2018), which improves upon the original model (Noah LSM v3.4.1) in several key ways. These modifications include an improved thermodynamic roughness length parameterization for sparse vegetation (Rodell et al., 2004) to correct the underestimation of ground heat flux, a new thermal conductivity scheme (Côté and Konrad, 2005) better suited for the coarse-grained, high-porosity soils common on the QTP, and an impedance factor related to ground ice content, which constrains the soil hydraulic conductivity to account for the impedance of water flow by ice (Zhang et al., 2007). The model's soil column was extended to a depth beyond the zero annual amplitude (~10 m for typical permafrost on the QTP (Zhao et al., 2010)) and discretized into multiple, vertically heterogeneous soil layers. This modified Noah LSM has been successfully validated at the Tanggula (TGL) site and applied in previous studies of permafrost degradation on the QTP (Ji et al., 2022; Zhang et al., 2022a), confirming its robust simulation capabilities in permafrost environment.

## 2.2 Implementation of NoahPy

The implementation of NoahPy involves recasting the numerical solution of the modified Noah LSM's governing equations into a fully differentiable structure. We use the following partial differential equations (PDEs) set to describe the dynamic system of the modified Noah LSM:

$$\begin{cases} \frac{\partial}{\partial t} s(t,z) = F(s(t,z), u(t,z), \beta^F) \\ y(t,z) = G(s(t,z), u(t,z), \beta^G) \end{cases} \tag{6}$$

where, $s(t,z)$ represents the state vectors that vary in time $t$ and space $z$ (e.g. soil temperature profile), $u(t,z)$ is the input vector of external forcings (e.g., meteorological data), $y(t,z)$ is the output vector (e.g., simulated variables for validation).

In the Noah LSM, the heat conduction (Equation 1) and Richards' (Equation 3) equations are solved using a finite-difference numerical approach. Following the spatial discretization scheme of Pan and Mahrt (1987) and the temporal scheme of Kalnay and Kanamitsu (1988), the PDEs are expressed in terms of explicit coefficients and implicit states. After discretization, the PDEs can be converted into a system of algebraic equations, which is then efficiently solved using the tridiagonal matrix algorithm. To ensure numerical stability, this calculation is applied twice for each time step when infiltration fluxes are large (Zheng et al., 2015).

The discretized form of Richards' equation, for example, for each soil layer *k* and time step *t* is:



$$\frac{\theta_k^{t+1} - \theta_k^t}{\Delta t} = \frac{1}{\Delta z_k}\left[ D(\theta_{k-1})\frac{\theta_{k-1}^{t+1} - \theta_k^{t+1}}{\widetilde{\Delta z_{k-1}}} - D(\theta_k)\frac{\theta_k^{t+1} - \theta_{k+1}^{t+1}}{\widetilde{\Delta z_k}} + K_{k-1} - K_k + S \right] \quad (7)$$

By letting $A = -\frac{D(\theta_{k-1})\Delta t}{\Delta z_k \widetilde{\Delta z_{k-1}}}, C = -\frac{D(\theta_k)\Delta t}{\Delta z_k \widetilde{\Delta z_k}}$, Equation 7 can be rearranged to:

$$A(\theta_{k-1}^{t+1} - \theta_{k-1}^t) + B(\theta_k^{t+1} - \theta_k^t) + C(\theta_{k+1}^{t+1} - \theta_{k+1}^t) = RHS \quad (8)$$

$$RHS = \frac{S + K_{k-1} - K_k}{\Delta z_k} \cdot \Delta t + A(\theta_k^t - \theta_{k-1}^t) + C(\theta_k^t - \theta_{k+1}^t), B = 1 - (A + C) \quad (9)$$

where $\Delta z_k$ is the thickness of the $k$-th soil layer; and $\Delta \widetilde{z_k}$ is the distance between the centers of layer $k$ and layer $k+1$. This equation can be rearranged into a tridiagonal system of linear equations, which is solved at each time step to update the soil

moisture profile, $\theta^{t+1}$:

$$\begin{bmatrix} B_1 & C_1 & 0 & 0 & 0 & \cdots & 0 \\ A_2 & B_2 & C_2 & 0 & 0 & \cdots & 0 \\ 0 & A_3 & B_3 & C_3 & 0 & \cdots & 0 \\ \vdots & \vdots & \vdots & \vdots & \vdots & \cdots & 0 \\ 0 & \cdots & 0 & A_{k-2} & B_{k-2} & C_{k-2} & 0 \\ 0 & \cdots & 0 & 0 & A_{k-1} & B_{k-1} & C_{k-1} \\ 0 & \cdots & 0 & 0 & 0 & A_k & B_k \end{bmatrix} \begin{bmatrix} \theta_1^{t+1} - \theta_1^t \\ \theta_2^{t+1} - \theta_2^t \\ \theta_3^{t+1} - \theta_3^t \\ \vdots \\ \theta_{k-2}^{t+1} - \theta_{k-2}^t \\ \theta_{k-1}^{t+1} - \theta_{k-1}^t \\ \theta_k^{t+1} - \theta_k^t \end{bmatrix} = \begin{bmatrix} RHS_1 \\ RHS_2 \\ RHS_3 \\ \vdots \\ RHS_{k-2} \\ RHS_{k-1} \\ RHS_k \end{bmatrix} \quad (10)$$

To make this process differentiable, we implemented the model within a RNN framework. A standard RNN updates an abstract hidden state, $h_t$, using a learned function (Figure 1a):

$$h_t = \sigma\left( W_h h_{(t-1)} + W_x x_t + b_h \right) \quad (11)$$

where $\sigma$ is the nonlinear activation function; $h_{t-1}$ and $h_t$ are the hidden states at the previous and current time steps, respectively; $W_h$ and $W_x$ are the weight matrices applied to the previous hidden state and the current input vector $x_t$, respectively; and $b_h$ is the bias vector.

In NoahPy, we replace this learned function with the entire physical time-step solution described above. The state of the system is a vector of physically meaningful variables, $s_t$ (e.g., soil temperature, moisture), which is updated according to the

model's deterministic physics (Figure 1b):

$$s_t = F_{Noah\,LSM}\left( s_{t-1}, x_t, \overrightarrow{\beta} \right) \quad (12)$$

where $F_{Noah\,LSM}$ represents the complete numerical solution for one time step, including the differentiable solver for the tridiagonal system derived from Equation 6; $x_t$ is the meteorological forcing, and $\overrightarrow{\beta}$ is the set of model parameters. This is made possible by implementing every step of the numerical solution using the differentiable operations native to the

PyTorch deep learning library (Paszke et al., 2019).



By constructing the model in this way, the entire time-stepping simulation allows the gradient of any model output with respect to any parameter ($\beta$) to be calculated efficiently using the backpropagation through time (BPTT) (Werbos, 1990), powered by PyTorch's automatic differentiation engine. Furthermore, all operations in NoahPy are vectorized to maximize the parallel computing power of modern hardware.

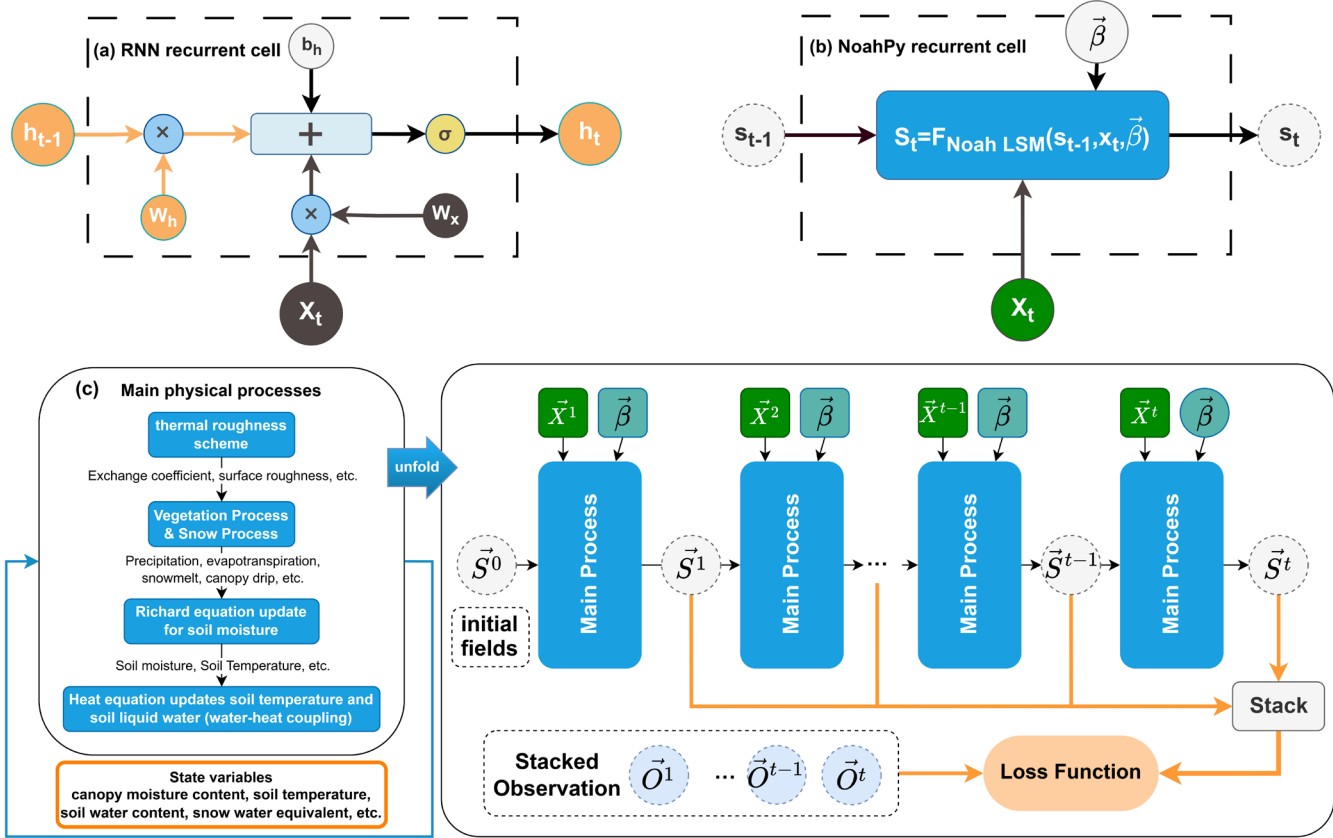


**Figure 1.** NoahPy architecture as a physics-based Recurrent Neural Network (RNN). (a) A standard RNN recurrent cell; (b) The NoahPy recurrent cell, which replaces the learned transformation with the physical model ($F_{\text{Noah LSM}}$); (c) The unfolded representation of the NoahPy simulation, where the model state (S) is updated at each time step. $\vec{X}$, $\vec{S}$, and $\vec{O}$ represent the meteorological forcing, state, and observation vectors, respectively, and $\vec{\beta}$ is the vector of model parameters.

## 2.3 Validations

### 2.3.1 Validation of numerical equivalence

The first validation step was to confirm that NoahPy, written in Python, accurately reproduces the numerical output of the original Fortran-based modified Noah LSM. This benchmark test ensures that the model rewriting





process did not introduce numerical artifacts. The experiment was conducted at three randomly selected grid cells on the QTP: Grid1 (28.75°N, 93.85°E), Grid2 (34.75°N, 98.25°E) and Grid3 (37.55°N, 100.55°E). Both models were driven by the China Meteorological Forcing Dataset (ITP-forcing) (He et al., 2020) for the period of 2000-2010. The year 1999 was used as a spin-up period (repeating for 500 years) to allow the model to reach equilibrium, and the model states at the end of this period were used as the initial conditions for the formal simulation. For both models, soil types were defined using the MSTD dataset (Wu and Nan, 2016), and vegetation types were based on the 1:1,000,000 China Vegetation Type Map (Zhang, 2007). Since the goal was a direct numerical comparison, model parameters were assigned using the default lookup table values corresponding to the soil and vegetation types. The soil column was configured with 18 layers extending to a depth of 15.2 m.

To quantify the agreement between the two models, we used three statistical metrics: Bias, Pearson correlation coefficient (R), and the Nash-Sutcliffe Efficiency coefficient (NSE):

$$\text{Bias} = \frac{1}{N}\sum_{i=1}^{N}\left(y_i - y_i^*\right) \tag{13}$$

$$R = \frac{\sum_{i=1}^{N}\left[(y_i - \bar{y})(y_i^* - \overline{y^*})\right]}{\sqrt{\sum_{i=1}^{N}\left[(y_i - \bar{y})^2\right]}\sqrt{\sum_{i=1}^{N}\left[\left(y_i^* - \overline{y^*}\right)^2\right]}} \tag{14}$$

$$\text{NSE} = 1 - \frac{\sum_{i=1}^{N}\left[(y_i - \bar{y})^2\right]}{\sum_{i=1}^{N}\left[\left(y_i^* - \overline{y^*}\right)^2\right]} \tag{15}$$

where, $y_i$ is a value from the NoahPy simulation time series, $y_i^*$ is the corresponding value from the modified Noah LSM simulation, $\bar{y}$ and $\overline{y^*}$ are the mean values of their respective time series, and N is the total number of samples.

**2.3.2 Validation of backpropagation capability**

To validate NoahPy's capability for backpropagation-driven parameter optimization, we conducted an experiment using observational data from the TGL permafrost site on the QTP. The model was driven by daily meteorological observations from the TGL station from April 1, 2007 to December 31, 2010. These data included air temperature, wind speed, relative humidity, incoming shortwave and longwave radiation, and precipitation. In-situ observations of active layer soil temperature and liquid water content from the site were used to constrain the model during optimization. The dataset was split into a training period (April 1, 2007 to December 31, 2009) and a validation period (January 1, 2010 to December 31, 2010). The NoahPy soil column was discretized into 20 layers to match the observation depths at the site. This included ten



shallow, higher-resolution layers (at 0.045, 0.091, 0.166, 0.289, 0.493, 0.829, 1.2, 1.6, 2.0, and 2.4 m) to capture rapid variations near the surface, and ten deeper layers (2.8, 3.8, 4.8, 5.8, 6.8, 7.8, 8.8, 10.8, 12.8, and 14.8 m) extending to 14.8 m.

The lower boundary of the simulation domain was set to a depth of 40 m, with the boundary temperature condition prescribed according to previous studies (Chen et al., 2015).

We selected four key soil hydraulic parameters, known to be highly sensitive to liquid water content (Brandhorst and Neuweiler, 2023; Szabó et al., 2024; Teuling et al., 2009), as the target for optimization: saturated hydraulic conductivity ($K_s$), saturated water content ($\theta_s$), soil matric potential at air entry ($\psi_s$), and the pore-size distribution index ($b$). The

allowable ranges for these parameters, drawn from previous studies (Rosero et al., 2009; Stuurop et al., 2021; Li et al., 2019; Wang et al., 2021), are provided in Table 1. Initial values were chosen randomly within these bounds. To ensure physical realism, we imposed a constraint that parameter values for the same soil type could not vary by more than 10% across different depths (Zhao et al., 2023).

The observational data for this study extend to a maximum depth of 2.45 m, corresponding to the model's 10th soil layer.

Therefore, simulated liquid water content from the top ten model layers was interpolated to the measurement depths. The NSE between the interpolated simulations and the observations was used as the loss function to be maximized. We used the widely adopted Adam optimizer (Kingma and Ba, 2014) with a learning rate of 0.0005 and default decay rates of 0.9 and 0.999.To improve convergence, a ReduceLROnPlateau learning rate scheduler was implemented. This scheduler monitored the NSE on the validation set and automatically reduced the learning rate by a factor of 0.1 if no improvement was observed

for ten consecutive epochs. The training was run for a maximum of 300 epochs, with a minimum learning rate of $1 \times 10^{-6}$ to prevent stagnation. The agreement between the optimized model simulations and the observations was quantified using the NSE, correlation coefficient, and Root Mean Square Error (RMSE).

**Table 1.** Target parameters to be optimized by backpropagation and their value ranges

| Parameter | Symbol (Unit) | Value Range |
|---|---|---|
| Saturated hydraulic conductivity | $K_s$ (m·s$^{-1}$) | $10^{-7}$-$6\times10^{-3}$ |
| Saturated water content | $\theta_s$ (m$^3$·m$^{-3}$) | 0.3-0.65 |
| Soil matric potential at air entry | $\psi_s$ (m) | 0.01-0.65 |
| Pore-size distribution | $b$ (Dimensionless) | 2.5-12 |

### 2.3.3 Performance comparison with traditional optimization

To demonstrate the advantages of a differentiable modeling approach, we compared the performance of NoahPy against both the original and modified Noah LSMs when calibrated with a traditional, widely used optimization algorithm. We evaluated three distinct model-optimizer combinations: NoahPy optimized with the gradient-based Adam optimizer; the



modified Noah LSM calibrated with the Shuffled Complex Evolution (SCE-UA) algorithm (Duan et al., 1994) ; and the original Noah LSM (v3.4.1) calibrated with the SCE-UA algorithm. The model configurations, forcing data, and target

parameters for all three setups were identical to those described in Section 2.3.2. A key difference is that the original Noah LSM does not account for vertical soil heterogeneity; therefore, its soil profile was configured uniformly using the properties of the surface layer. For a robust comparison, each optimization algorithm was run ten times with a maximum of 500 iterations.

In addition to the NSE, we used the Kling-Gupta Efficiency (KGE) as a more comprehensive performance metric. KGE

provides a multi-faceted assessment by decomposing performance into three distinct components:

$$KGE = 1 - \sqrt{\left( Corr - 1 \right)^2 + \left( \alpha - 1 \right)^2 + \left( \gamma - 1 \right)^2} \tag{16}$$

where, $Corr$ is the Pearson correlation coefficient between simulated and observed values, $\alpha$ is the bias ratio (mean of simulated values / mean of observed values), and $\gamma$ is the variability ratio (coefficient of variation of simulated values / coefficient of variation of observed values). To determine if the performance differences among the three model setups were

statistically significant, we employed a two-step non-parametric testing procedure on the KGE values from all soil depths. First, the Friedman test was used to assess whether any significant differences existed within the group of three models. If the Friedman test returned a p-value < 0.05, we then performed the Dunn's post-hoc test for pairwise comparisons to identify which specific model pairs differed significantly from one another. A p-value < 0.05 in the Dunn's test was considered a statistically significant difference in performance.

**3 Results**

**3.1 Numerical equivalence with the modified Noah LSM**

The validation confirms that NoahPy successfully replicates the numerical behaviour of the Fortran-based modified Noah LSM. As shown in the scatter plots in Figure 2, the simulated daily soil temperature and liquid water content from NoahPy exhibit a near-perfect 1:1 relationship with the outputs from the modified LSM across all tested depths (0.1, 0.5, 0.8,

1.3, and 2.5 m) aggregated from three randomly chosen grid cells on the QTP. The performance is exceptionally strong, with NSE coefficients greater than 0.999 and near-zero bias (<0.01) for both variables at every depth.

A minor degree of scatter is visible in the soil moisture comparisons (Figure 2b, d, f, h, j), which is not present in the soil temperature results. These small deviations are likely attributable to minor differences in floating-point arithmetic and numerical precision between the Python/PyTorch environment and the original Fortran compiler. Importantly, NoahPy

maintains this high accuracy in deeper soil layers, with no amplification of numerical errors with depth. This demonstrates the high numerical stability of the NoahPy implementation and confirms that it serves as a faithful and reliable replacement for the original model.



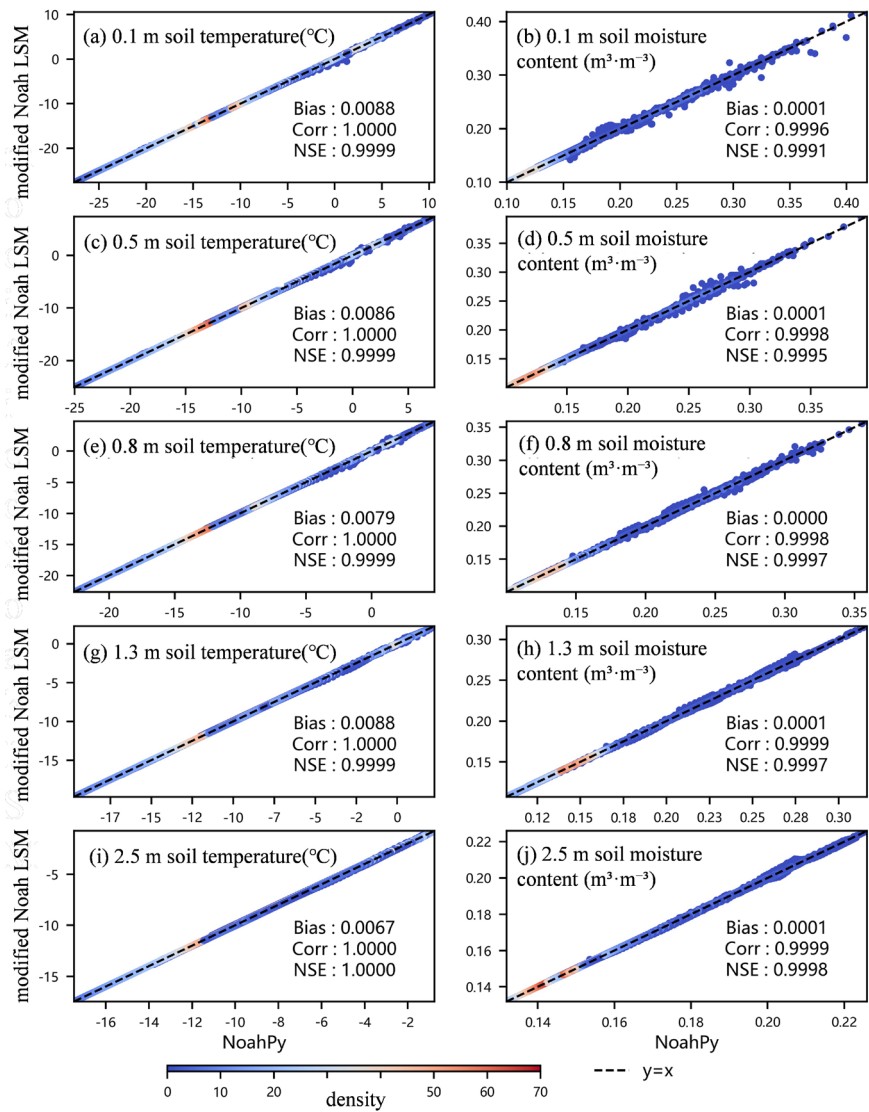

**Figure 2.** Comparison of NoahPy and modified Noah LSM outputs for soil temperature and moisture. The density scatter plots compare daily model outputs at five different soil depths (0.1, 0.5, 0.8, 1.3, and 2.5 m), aggregated from three randomly chosen grid cells (28.75°N, 93.85°E; 34.75°N, 98.25°E; 37.55°N, 100.55°E) on the Tibetan Plateau (QTP). The dashed line represents perfect agreement (y=x). Inset values show the Bias, correlation (Corr), and NSE.

## 3.2 Performance of the calibrated NoahPy at the Tanggula site

The gradient-based optimization process effectively calibrated the NoahPy model parameters. The training process demonstrates rapid convergence, with the NSE for soil liquid water increasing from an initial value of -0.2 to an optimal value of 0.84 (Figure 3). Correspondingly, the RMSE steadily decreases. This result successfully validates that NoahPy's



differentiable framework allows for the effective use of backpropagation to optimize model parameters against observational data.


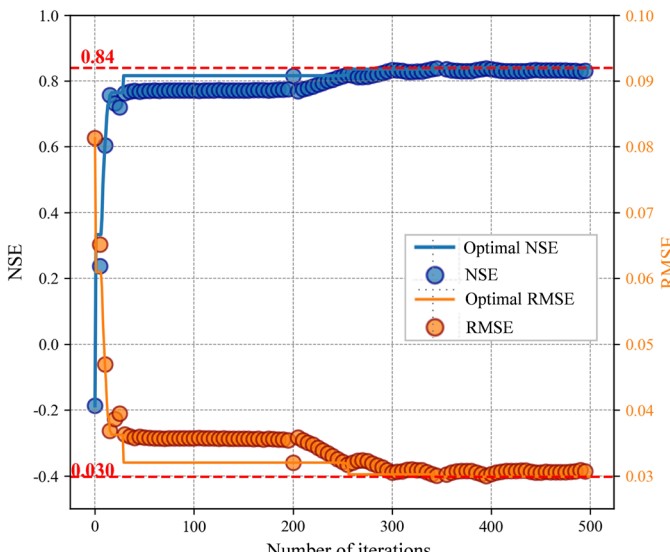

**Figure 3.** Training convergence for the soil liquid water simulation at the Tanggula site. The plot shows the improvement in the Nash-Sutcliffe Efficiency (NSE, blue line) and the corresponding reduction in the Root Mean Square Error (RMSE, orange line) over 500 optimization iterations. The dashed red lines mark the best performance achieved.

After calibration, NoahPy's simulations showed excellent agreement with the observed data at the TGL site during both the calibration (2007-2009) and validation (2010) periods (Figure 4). The model accurately reproduced the seasonal cycle of soil temperature at all depths. For most layers, the NSE values exceeded 0.9, and the RMSE decreased with depth, reflecting the reduced temperature variability in deeper soil. However, the model exhibits a cold bias during the winter of 2008–2009, with simulated temperatures falling below observations (Figure 4a). This period was characterized by heavy snowfall at the

site. The cold bias is likely due to the relatively simplistic snow scheme in the Noah LSM, which can underestimate snow depth. A shallower simulated snowpack provides less insulation, allowing excessive heat loss from the soil to the cold atmosphere. Additionally, anomalous fluctuations were observed in the measured deep soil temperatures (1.05 m and 2.45 m) during the summer of 2009 (Figure 4d, e). Given that deep soil temperatures should respond slowly to short-term atmospheric changes, these fluctuations are likely attributable to instrumental error.

While more complex than temperature, the dynamics of soil liquid water were also well-captured, with NSE values exceeding 0.7 and RMSE values below 0.05 $m^3\ m^{-3}$ for most layers. The model successfully simulated soil moisture responses to freeze-thaw cycles and summer precipitation events, particularly in the shallow soil layers (Figure 4f, g). However, several discrepancies were noted, particularly in deeper soil. Simulations at depths of 1.05 m and 2.45 m deviate



more pronouncedly from the measured data (Figure 4i, j). The model tended to overestimate liquid water content during the

winter freezing period at some depths (Figure 4h, i). This can be attributed to the model's hydraulic parameterization scheme, which is based on the Campbell formulation; this approach neglects the effects of ice suction and effective porosity. Omitting these mechanisms, which influence soil water redistribution at the freezing front, can lead to an overestimation of liquid water content during winter (Zhao et al., 2023). Additionally, some observations appear anomalous. For example, the measured unfrozen water content in winter drops to exactly zero at 0.4 m and 1.05 m, which is physically unlikely and

suggests potential instrument error at low moisture levels. Similarly, sharp, isolated increases in measured water content at deeper layers during the summer of 2009 (Figure 4h, i, j) without corresponding signals in the layers above suggest these are likely not caused by surface infiltration and may also be data artifacts.

       Despite the well-diagnosed limitations of specific model parameterizations and potential artifacts in the observational data, the results for all soil depths demonstrate that the calibrated NoahPy model reliably reproduces the key seasonal

dynamics of soil temperature and liquid water during complex freeze-thaw cycles at the TGL site.







**Figure 4.** Simulated and observed daily soil temperature and liquid soil water content at various depths (daily; 0.05, 0.1, 0.4, 1.05, and 2.45m) for the Tanggula (TGL) site. The vertical black dashed line separates the calibration period (April 1, 2007–December 31, 2009) from the validation period (January 1, 2010–December 31, 2010). Inset text in each panel provides the NSE, RMSE, and correlation coefficient (Corr) for both periods.





### 3.3 Comparative performance evaluation

The primary advantage of the differentiable approach is evident in the parameter optimization process. NoahPy paired with the Adam optimizer converges extremely rapidly, reaching a high level of accuracy within only 100 iterations (Figure 5). This is due to the Adam optimizer's use of gradient information and an adaptive learning rate. In contrast, the traditional

SCE-UA algorithm applied to the Noah and modified Noah LSMs converges much more slowly, requiring significantly more iterations to approach an optimal solution (Figure 5). While the SCE-UA algorithm's strength is its global search capability, which helps it avoid getting trapped in local optima, its convergence becomes prohibitively slow in high-dimensional parameter spaces, requiring significantly more iterations to find a solution. Furthermore, the gradient-based approach demonstrates greater stability. The shaded 95% uncertainty band around the convergence trajectory for NoahPy is

visibly narrower than for the SCE-UA method ((Figure 5), indicating that the Adam optimizer finds a robust solution more consistently across repeated runs.

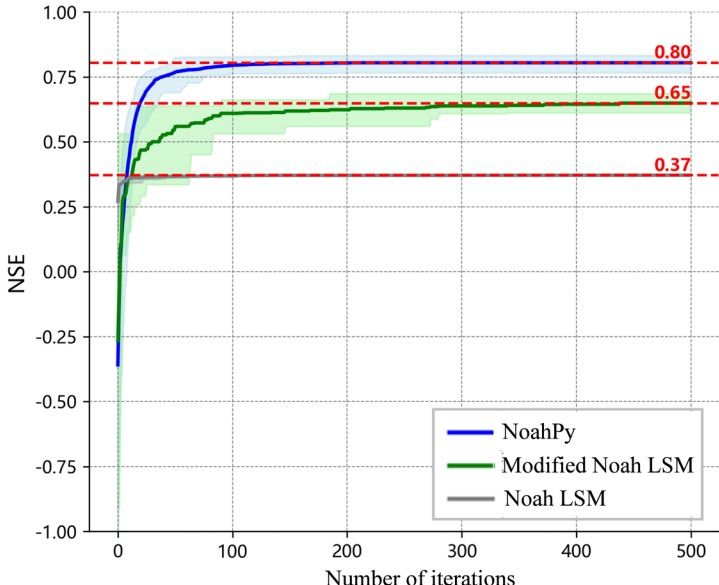

**Figure 5.** Convergence of NoahPy, the modified Noah LSM, and the original Noah LSM in terms of NSE. Each line represents the mean NSE from 10 optimization runs, with the shaded area indicating the 95% uncertainty band. NoahPy was optimized with the Adam

optimizer, while the other two models were calibrated with the SCE-UA algorithm.

When comparing the calibrated models' ability to simulate soil temperature ((Figure 6), all three setups perform well in the shallow soil layers (0.05 m and 0.4 m), with NSE values exceeding 0.9. However, a major performance gap appears in the deep soil (2.45 m). The original Noah LSM, which neglects vertical soil heterogeneity, exhibits a pronounced cold bias, with an RMSE of 1.68°C (Figure 6i). NoahPy and the modified Noah LSM, which both account for varying soil layers,

perform significantly better, with RMSE values of 0.51°C (Figure 6c) and 0.85°C (Figure 6f), respectively. In essence, the



error is magnified with depth because the impact of incorrect thermal properties is compounded over the longer time and distance it takes for heat to travel to the deep soil.

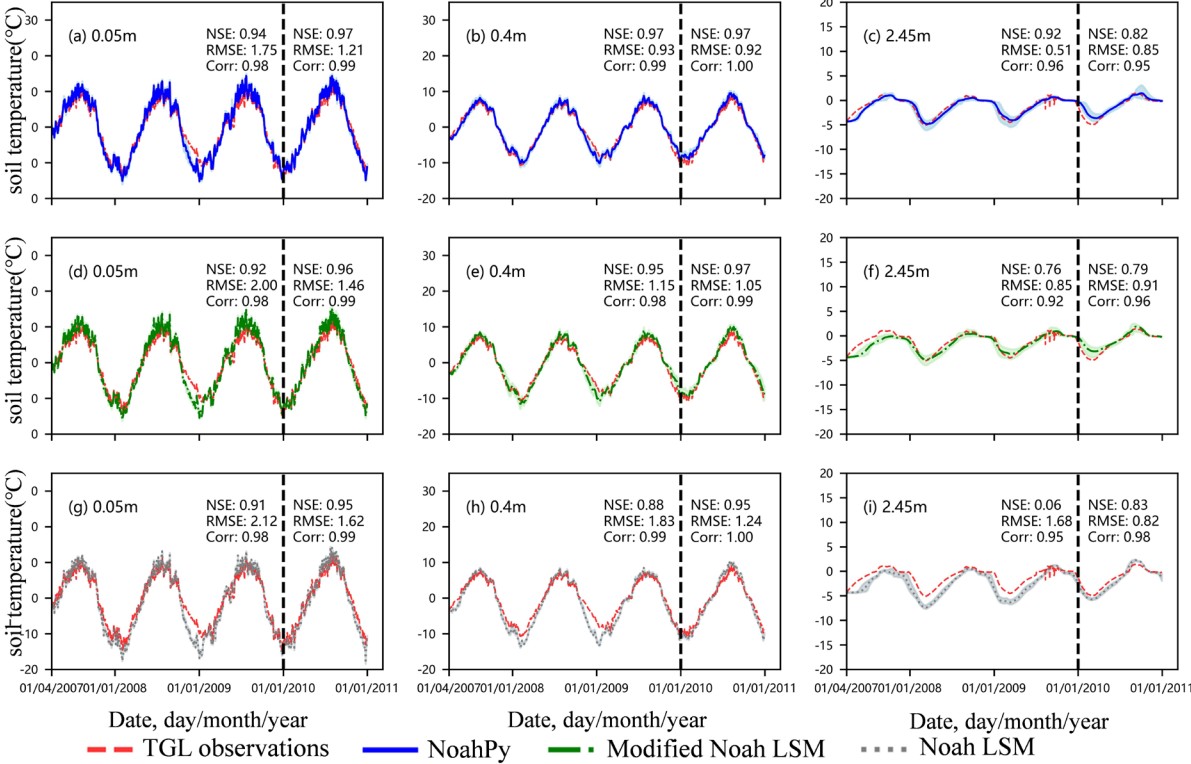

**Figure 6.** Comparison of calibrated model performance for daily soil temperature at the TGL site. Each panel compares in-situ observations (red dashed line) against the simulations from the three calibrated models at a specific depth. The models are NoahPy (optimized with Adam; blue line and shading), the modified Noah LSM (calibrated with SCE-UA; green line and shading), and the original Noah LSM (calibrated with SCE-UA; gray dashed line and shading). The shaded areas represent the 95% uncertainty band from 10 repeated optimization runs. The vertical dashed line separates the calibration and validation periods.

The results for soil liquid water simulation show an even starker contrast (Figure 7). Both NoahPy and the modified Noah LSM produce satisfactory results, with RMSE below 0.05 m³ m⁻³ across all three layers. These models accurately capture the key seasonal dynamics, including soil moisture fluctuations driven by summer precipitation and the rapid changes associated with freeze-thaw phase transitions, which align well with observations. The original Noah LSM, however, performs poorly. It fails to capture moisture fluctuations from summer rainfall and shows significant biases in winter. Its performance deteriorates sharply with depth, with the NSE value dropping to -0.09 in the deepest layer (2.45 m) (Figure 7i). This negative NSE reflects a substantial underestimation of the liquid water increase during the spring thaw. This finding is consistent with previous research (Wu et al., 2018).



A Friedman test performed on the KGE values for all models (Table 2) confirmed a statistically significant difference in their overall performance (p ≈ 0). A subsequent Dunn's post-hoc test revealed that both NoahPy and the modified Noah LSM performed significantly better than the original Noah LSM. Interestingly, the statistical test showed no significant
difference between NoahPy and the modified Noah LSM (p = 0.1659). This is expected, as they share identical physics. However, NoahPy consistently demonstrated practical advantages in performance. As shown in Figure 5, NoahPy converges markedly faster with the Adam optimizer, approaching its optimal solution in roughly 100 iterations, whereas the modified Noah LSM requires substantially more iterations to converge with the SCE-UA algorithm. Furthermore, NoahPy's final calibrated simulations have noticeably lower uncertainty (i.e., smaller shaded bands in Figures 6 and 7) compared to the
modified Noah LSM, particularly for winter liquid water content (Figure 7a,c vs. (Figure 7d,f). This lower uncertainty is a direct result of the more stable and efficient optimization provided by the gradient-based Adam algorithm, highlighting a key practical advantage of the differentiable modeling approach.

**Table 2.** Mean Kling-Gupta Efficiency (KGE) values for the three calibrated models. Values represent the mean KGE from 10 repeated optimization runs for NoahPy, the modified Noah LSM, and the original Noah LSM.

| Variable | Depth(m) | NoahPy | Modified Noah LSM | original Noah LSM |
|---|---|---|---|---|
| Soil temperature | 0.05 | 0.83 | 0.79 | 0.79 |
|  | 0.1 | 0.86 | 0.81 | 0.8 |
|  | 0.4 | 0.93 | 0.91 | 0.74 |
|  | 1.05 | 0.89 | 0.85 | 0.57 |
|  | 2.45 | 0.93 | 0.83 | 0.28 |
| Soil liquid water content | 0.05 | 0.91 | 0.82 | 0.51 |
|  | 0.1 | 0.95 | 0.88 | 0.69 |
|  | 0.4 | 0.64 | 0.51 | 0.23 |
|  | 1.05 | 0.64 | 0.52 | 0.31 |
|  | 2.45 | 0.8 | 0.56 | 0.27 |



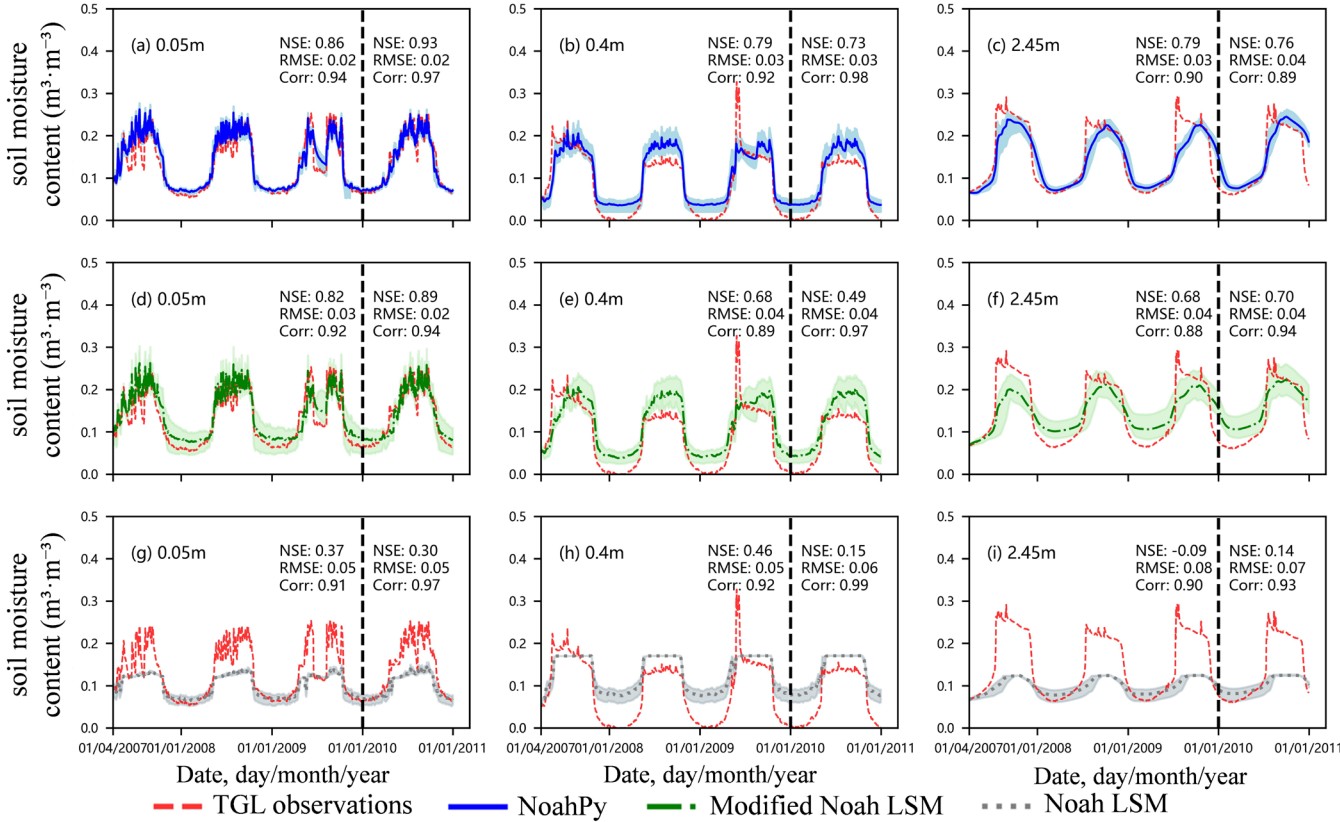

**Figure 7.** Comparison of calibrated model performance for daily liquid soil water at the TGL site. Each panel compares in-situ observations (red dashed line) against the simulations from the three calibrated models at a specific depth. The models are NoahPy (optimized with Adam; blue line and shading), the modified Noah LSM (calibrated with SCE-UA; green line and shading), and the original Noah LSM (calibrated with SCE-UA; gray dashed line and shading). The shaded areas represent the 95% uncertainty band from 10 repeated optimization runs. The vertical dashed line separates the calibration and validation periods.

## 4 Discussion

This study successfully demonstrated the development and application of NoahPy, a fully differentiable land surface model for permafrost. Our results confirm that this re-implementation not only preserves the physical integrity of the modified Noah LSM but also unlocks a parameter optimization workflow that is significantly faster and more robust than traditional methods. The successful calibration and diagnostic analysis in this study highlight the theoretical merits of our "glass-box" approach. A common alternative for making a physical model compatible with machine learning workflows is to develop a surrogate model: a neural network trained to mimic the input-output behavior of the original, non-differentiable code (Razavi et al., 2012). While easier to implement, this approach treats the model as a "black box" and suffers from the



curse of dimensionality (Asher et al., 2015). As the number of parameters grows, the required simulations increase
exponentially, making surrogates infeasible for complex LSMs. While such a surrogate could potentially replicate the final
simulation results, it obscures the internal model dynamics. In contrast, the full interpretability of NoahPy allowed us to
diagnose specific physical process errors, such as the cold bias from the simplified snow scheme and the overestimation of
winter liquid water due to missing cryosuction physics. This ability to directly attribute simulation errors to specific physical
parameterizations is a fundamental advantage of the differentiable physics-based approach and is essential for targeted
scientific model improvement.

Gradient-based optimization is particularly advantageous when coupling NoahPy with neural networks for hybrid
modeling. It allows the simultaneous calibration of a large number of model parameters, which would be prohibitively
difficult using traditional gradient-free methods such as SCE-UA. While SCE-UA can perform a global search and avoid
local minima, its performance degrades substantially in high-dimensional parameter spaces. By contrast, optimizers like
Adam exploit precise gradients to iteratively improve parameter values, facilitating effective end-to-end training of hybrid
systems. It should be noted that we do not provide absolute comparisons of computational speed, as differences in model
implementation (Fortran vs Python) and numerical schemes limit direct benchmarking. Instead, the focus here is on the
iterative optimization capability of gradient-based methods, which underpins the scalability and feasibility of hybrid training
strategies.

This framework holds promise for addressing challenges in permafrost domain, where parameterization for key soil
properties in permafrost environment such as Qinghai-Tibet Plateau (QTP) like thermal conductivity (Ji et al., 2024),
hydraulic conductivity (Hu et al., 2023), and matric potential (Zhao et al., 2023) may be incomplete. While NoahPy, in its
current form, inherits the physical limitations of its parent model, its true power lies in its potential as a foundational
framework for a new generation of hybrid models.The NoahPy framework allows for coupling with external machine
learning models that can learn the complex mapping between environmental covariates (e.g., topography, vegetation, soil
type) and the model's physical parameters (such as hydraulic and thermal parameters) from direct observations (e.g., soil
temperature, soil moisture content). This could dramatically improve the spatial transferability of parameters across diverse
regions, reducing the reliance on costly site-specific calibration and mitigating parameter uncertainty, a key challenge in
permafrost modelling (Harp et al., 2016; Dai et al., 2019). The hybrid, seamless physics-machine learning models coupling
enabled by automatic differentiation also allows for targeted replacement of model components. For instance, empirical
parameterizations where physical knowledge is weak, such as the Campbell-based hydraulic scheme, can be replaced by an
embedded neural network. In such a hybrid mode, the neural network can learn more complex and accurate relationships
from data, while the surrounding physical equations ensure its predictions remain constrained by fundamental laws like the
conservation of mass and energy.

This study has two primary limitations. First, while successfully validated at the Tanggula site on the Qinghai-Tibet
Plateau, the performance and applicability of NoahPy in other permafrost regions with different characteristics (e.g., the ice-





rich Yedoma of Siberia or the boreal forests of North America) have yet to be confirmed. Second, NoahPy inherits the known physical deficiencies of the Noah LSM, including a simplistic snow scheme and the omission of processes critical to permafrost carbon cycling, such as the effects of soil organic matter, convective heat transfer, and abrupt thaw dynamics.

The framework presented here is not intended as a final product, but as a flexible and extensible foundation for the community. By recasting a widely-used LSM into the deep learning ecosystem, we have created a tool that can leverage the rapid advancements in computational hardware (e.g., GPUs, TPUs) and software (Sevilla et al., 2022; Kochkov et al., 2024). This work helps bridge the gap between process-based modeling and AI, establishing a path toward the next generation of hybrid Earth System Models capable of reducing uncertainty and providing more reliable projections of the future of the

cryosphere.

## 5 Conclusions

In this study, we developed NoahPy, a fully differentiable land surface model specifically improved for permafrost thermo-hydrology. We successfully recast the widely-used, Fortran-based Noah LSM into a "glass-box" Python framework that is both physically interpretable and fully compatible with gradient-based optimization. Based on our results, we draw the

following key conclusions:

(1) NoahPy perfectly reproduces the numerical behaviour of the modified Noah LSM. Validations show a near-perfect match, with NSE values exceeding 0.99 for both soil temperature and liquid water across all soil layers, confirming the fidelity of the model's re-implementation.

(2) The differentiable framework enables robust, gradient-based parameter optimization. Validation at a permafrost site

on the QTP demonstrates that NoahPy can effectively use backpropagation to learn from observational data. The resulting calibrated model shows strong performance, achieving NSE values above 0.9 for soil temperature and 0.8 for liquid water.

(3) The NoahPy-Adam workflow is superior to traditional calibration methods. The combination of the differentiable model with a gradient-based optimizer (Adam) results in a parameter optimization that is significantly faster, more stable, and yields final simulations with lower uncertainty compared to the traditional SCE-UA algorithm.

This work delivers a foundational tool that was previously missing for the permafrost community. It closes the technical gap that has hindered the development of deeply-integrated hybrid models for the cryosphere. This study thus lays the necessary groundwork for future AI-based models that aim to lower uncertainty and deliver more credible predictions of permafrost's response to a changing climate.

*Code and data availability.* The NoahPy model code used in this study is available at https://github.com/nanzt/NoahPy, and the

exact version used to generate the results presented here is archived on Zenodo (Tian and Nan, 2025b, https://doi.org/10.5281/zenodo.16530326). The original Noah LSM (v3.4.1) code used in this study is available at https://ral.ucar.edu/model/unified-noah-lsm (last access: August 28, 2025). The modified version of Noah LSM code is



available at https://doi.org/10.17605/osf.io/g7jqr (Zhang et al., 2022b). The simulation data generated in this study are available on Figshare (Tian and Nan, 2025a, https://doi.org/10.6084/m9.figshare.29988163).

*Author contributions*. **Wenbiao Tian:** Formal analysis, Investigation, Methodology, Software, Validation, Writing – original draft, Writing – review & editing. **Hu Yu**: Formal analysis, Validation. **Shuping Zhao:** Funding acquisition, Supervision, Writing – original draft, Writing – review & editing. **Yuhe Cao**: Formal analysis. **Wenjun Yi**: Investigation, Writing – original draft, Writing – review & editing. **Jiwei Xu**: Writing – original draft, Writing – review & editing. **Zhuotong Nan:** Conceptualization, Funding acquisition, Methodology, Resources, Writing – original draft, Writing – review & editing.

*Competing interests*. The contact author has declared that none of the authors has any competing interests.

*Acknowledgements*: This work is supported by the National Key Research and Development Program of China (No. 2022YFF0711703) and National Natural Science Foundation of China (No. 42171125, 42571149).

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
