# Peer review of "NoahPy: A differentiable Noah land surface model for simulating permafrost thermo-hydrology"

_EGUsphere, 2025_

## Author Response (AR1)

**Response to the editor**

Dear Dr. Ebel,

Thank you for your notification and helpful suggestions.

We have carefully addressed both points as follows:

1. In the *Author Contribution* section, full author names have been replaced by their initials as requested.

2. We have carefully checked Figures 5, 6, and 7 using the Coblis – Color Blindness Simulator. The results indicate that the different models can still be easily distinguished under various types of colour vision deficiencies. Therefore, no further changes to the colour schemes were made, as the figures already ensure sufficient visual distinction through both colour and line style differences.

The revised manuscript has been uploaded accordingly.

**Response to Review Comment #1**

*Note: The line numbers mentioned in the reply refer to the line numbers in the change track document.*

The state-of-the-art land surface models (LSMs) have been reported to perform poorly in representing permafrost processes. To address this gap, the authors present NoahPy—a fully differentiable LSM developed by reconstructing the Noah LSM's governing partial differential equations into a process-encapsulated recurrent neural network. NoahPy was compared with both the original and an improved version of the Noah LSM, and evaluated at a permafrost site. I find the model to be skillful and the results reasonable.

Response:We sincerely thank the reviewer for the constructive and thoughtful comments, which helped us improve the quality and clarity of the manuscript. We have prepared revisions to address all of these comments.

1. Introduction: It would be beneficial to restructure the introduction to better highlight the significance of permafrost, particularly as the authors aim to introduce the model to the permafrost research community. The section could begin by underscoring the importance of permafrost, followed by a critical review of how current LSMs represent permafrost processes, clearly outlining existing limitations. Addressing this gap, the authors should then introduce deep learning methods and explain how such approaches can provide an effective solution to improve permafrost modeling.

Response: Thank you.

The reviewer suggested a restructuring for the Introduction section and introducing deep learning as a solution to improve permafrost modeling. While we agree on the importance of these topics, our paper's central contribution is slightly different. Our primary goal is not to introduce new physics to solve the well-documented LSM deficiencies, but rather to solve a critical technical gap: the non-differentiable nature of existing permafrost-capable LSMs.

This technical gap is what currently prevents the permafrost community from leveraging the power of hybrid AI modeling, AI-driven calibration, and end-to-end differentiable workflows. Therefore, our introduction is structured to first establish the promise of hybrid AI, then identify the differentiable gap, and then frame our work as the solution to this specific technical gap for the permafrost community.

In this revision, to better address the reviewer's points, we enhanced our permafrost-centric focus in the introduction section as well as other sections. For example, in the Introduction section:

"This challenge is especially pronounced in complex, data-scare environments like the cryosphere." (lines 32-33)

"The primary obstacle to this integration for the land surface and permafrost modeling community has been technical: most established geophysical models, including well-known land surface models (LSMs), are implemented as non-differentiable numerical solvers, making them incompatible with the gradient-based optimization central to deep learning (Rumelhart et al., 1986)." (lines 40-43)

"While significant effort has gone into improving the physics of permafrost specific models (Ji et al., 2022; Wu et al., 2018; Xiao et al., 2013; Zhao et al., 2023), these improved models remain non-differentiable, preventing their integration into model AI-driven calibration and hybrid modeling workflows." (lines 58-60)

"To address this gap, we introduce NoahPy: a fully differentiable land surface model specifically improved LSM built upon a version of the Noah LSM already modified and validated for simulating permafrost thermos-hydrology on the Qinghai-Tibet Plateau (QTP). We have rewritten the widely-used this permafrost centric, Fortran-based Noah LSM model into a differentiable Python framework by encapsulating its governing partial differential equations within a Recurrent Neural Network (RNN) structure." (lines 70-74)

In Conclusions section:

"NoahPy faithfully reproduces the numerical behaviour of the permafrost-specific modified Noah LSM. Validations show a very close match, with NSE values exceeding 0.99 for both soil temperature and liquid water across all soil layers, confirming the fidelity of the model's re-implementation." (lines 429-430)

Please refer to the revised manuscript for all the changes we have made.

2. Discussion: The advantages and limitations are currently intermingled in this section. Please consider: (a) adding a brief outlook on future model development; and (b) using subsections to enhance the readability of the manuscript.

Response:

We thank the reviewer for the careful reading and constructive comments on the "Discussion" section of our manuscript. We fully understand the two key issues raised: (a) the mixing of model strengths and limitations within the section, and (b) the lack of outlook on future model development. In response, while retaining the "Discussion" as a single cohesive section, we have reorganized its structure and logic as follows:

**1) Structural adjustments:**

Within the "Discussion" section, we have reordered the paragraphs to improve clarity and logical flow. We first highlight the main advantages of NoahPy, including the significant improvements brought by its differentiable framework in model transparency, parameter optimization efficiency, and error diagnostics, as well as its higher stability and scalability compared with traditional optimization algorithms such as SCE-UA.

Then, we discuss the current limitations of the model, such as inheriting known physical deficiencies from the Noah LSM and its validation being primarily limited to the TGL sites on the Tibetan Plateau.

Finally, we have added a paragraph outlining future model development. We emphasize that NoahPy is not a finished product but an open and extensible framework intended to provide the permafrost modeling community with a platform for continuous improvement. This framework supports deep coupling with external machine learning models and can learn complex mappings between environmental covariates (e.g., topography, vegetation, soil type) and physical parameters (e.g., hydraulic conductivity, thermal conductivity), thereby enhancing regional transferability of model parameters and reducing reliance on expensive pointwise calibration, effectively mitigating parameter uncertainty in permafrost simulations. Leveraging automatic differentiation, NoahPy also enables "modular" updates of specific physical processes, for example, embedding neural networks to replace empirical hydraulic parameterizations, while preserving energy and mass conservation constraints and learning more accurate physical relationships. This work helps bridge the gap between process-based modeling and AI, establishing a path toward the next generation of hybrid Earth System Models capable of

reducing uncertainty and providing more reliable projections for the future of the cryosphere.

**2) On sectioning:**

We carefully considered the suggestion to add subsection headings. However, given the relatively concise length of the "Discussion," splitting it into multiple subsections would result in overly short segments, potentially disrupting overall coherence and reading flow. Therefore, we opted to maintain a single-section structure, using natural transitions and logical connections to differentiate between model strengths, limitations, and future perspectives.

We believe that these revisions significantly improve the logical flow and readability of the discussion while fully addressing the reviewer's comments regarding structural clarity and outlook on future model developments.

Please refer to the revised Discussion section for all the changes we have made.

3. The language should be improved throughout for clarity and academic tone.

Response: Thanks, we have improved the language thoroughly in this revision.

4. L18: Avoid using the word "perfectly," as no model can be considered perfect. Please revise this throughout the manuscript (e.g., L224 and others). As noted on the GMD homepage:"Essentially, all models are wrong, but some are useful." (George E. P. Box, 1979)

Response:We thank the reviewer for this valuable suggestion. We agree that the word "perfectly" may imply unrealistic precision and have revised it throughout the manuscript. All "perfect" and similar words are revised. Specifically:

- Line 18 in the Abstract: we replaced "perfectly replicates" with "very closely replicates."
- Line 429 in the Conclusions: we replaced "perfectly reproduces" with "faithfully reproduces."

- Line 430 in the Conclusions: we changed "near-perfect match" to "very close match."

All similar expressions have been revised accordingly to avoid overstatement.

5. L22: "SCE-UA" is not defined.

Response: We thank the reviewer for the careful reading and valuable suggestion. We agree with this comment. The full name "Shuffled Complex Evolution" has been provided when it appears in the abstract (line 23).

6. L105: Eq.6: Clarify what "$\beta^F$" and "$\beta^G$" refer to.

Response:$\beta^F$ and $\beta^G$ represent parameter sets involved in the control equations and output equations, respectively. We explain them in line 117.

7. L204: While a reference is provided, please explain the principle of the Shuffled Complex Evolution (SCE-UA) algorithm. For example, what does the strength of the SCE-UA algorithm stem from?

Response:We thank the reviewer for this constructive comment. In the revised manuscript, we have added a brief explanation of the underlying principle and strengths of the Shuffled Complex Evolution (SCE-UA) algorithm in Section 2.3.3 (lines 229-237) :

"The SCE-UA algorithm (Duan et al., 1994) is a widely used global optimization method that combines probabilistic sampling with competitive evolution. It starts by generating multiple complexes, each representing a subgroup of candidate parameter sets. Within each complex, solutions evolve independently through processes analogous to selection, crossover, and mutation to produce new trial members. Periodic shuffling of complexes allows information exchange among subpopulations, helping the search escape local minima and preserve population diversity (Rahnamay Naeini et al., 2019) ; This shuffled and competitive framework enables SCE-UA to efficiently balance global exploration and local exploitation, offering strong robustness and reliability for calibrating complex, nonlinear hydrological and land surface models." (lines 229-237)

**References:**

Duan, Q., Sorooshian, S., and Gupta, V. K.: Optimal use of the SCE-UA
global optimization method for calibrating watershed models, Journal of
Hydrology, 158, 265-284, https://doi.org/10.1016/0022-1694(94)90057-4,
1994.

Rahnamay Naeini, M., Analui, B., Gupta, H. V., Duan, Q., and Sorooshian, S.:
Three decades of the Shuffled Complex Evolution (SCE-UA) optimization
algorithm: Review and applications, Scientia Iranica, 26, 2015-2031,
https://doi.org/10.24200/sci.2019.21500, 2019.

8. L255: This could be easily verified by additionally evaluating snow water
equivalent or snow depth against observations. I assume snow depth data are
available at the TGL site (e.g., Xiao et al., 2013).

Response:

As suggested, we obtained the daily snow depth observations from the TGL
site (with gratitude to the Cryosphere Research Station on the Qinghai–Tibet
Plateau, CAS) and compared them with our NoahPy simulations.

This comparison (Figure S1) provides definitive proof of our statement. Our
model significantly underestimates the peak snow depth during the 2008-2009
winter. This is precisely the period where our soil temperature simulation
exhibited its most pronounced cold bias (as seen in our Figure 4a). The
model's simulated snowpack is far too shallow and melts too quickly, which
confirms that the lack of insulation from this underestimated snowpack is the
potential cause of the simulated soil temperature bias.

We have revised the manuscript in Section 3.2 to reflect this new, stronger
evidence.

"However, the model exhibits a cold bias during the winter of 2008–2009, with
simulated temperatures falling below observations (Figure 4a). This period
was characterized by heavy snowfall at the site. The cold bias is confirmed to
be a direct result of the relatively simplistic snow scheme in the Noah LSM. A
direct comparison with observed snow depth data from the TGL site shows

the model significantly underestimates the peak snow accumulation during this exact 2008-2009 winter and melts the snowpack too rapidly. The resulting shallower simulated snowpack provides less insulation, allowing excessive heat loss from the soil to the cold atmosphere." (lines 286-291)

We are not adding this new figure to the main manuscript, as we wish to keep the paper's focus squarely on the soil thermo-hydrology and the novel differentiable workflow. However, we are very grateful for this suggestion

[Figure]

Figure S1 Comparison of Simulated(NoahPy) and Observed Snow Depth

9. L296: There are two opening parentheses here. Similar typos occur elsewhere in the manuscript; please revise carefully.
Response: The typos have been corrected in the revised manuscript.

10. L407: What does the underline signify here?
Response: The underline has been removed in the revised manuscript.

**Response to Review Comment #2**

*Note: The line numbers mentioned in the reply refer to the line numbers in the change track document.*

Tian et al. present NoahPy, a differentiable reformulation of the Noah land surface model (LSM) aimed at improving the representation of permafrost

thermo-hydrology. The authors rewrite the traditional Fortran-based Noah LSM into a PyTorch-based, partially differentiable framework and demonstrate that it reproduces the original model's numerical behavior while enabling gradient-based parameter optimization through backpropagation.

Strengths

1. The paper provides a clear and rigorous implementation of a differentiable land surface model using PyTorch.
2. The model reproduces the original Fortran Noah LSM with high fidelity (NSE > 0.99), indicating numerical equivalence.
3. The differentiable structure allows efficient gradient-based calibration (using Adam), showing faster and more stable convergence than traditional SCE-UA optimization.
4. The manuscript is well organized, the methodology transparent, and the validation experiments are convincing for the scope of the technical contribution.

Response:We sincerely appreciate the reviewer's positive and encouraging comments.

Major comments

1. Scope of differentiability vs. the claim of "fully differentiable LSM". Although the abstract and conclusions describe NoahPy as a fully differentiable LSM, the actual implementation appears to make only the heat and moisture transport equations (the PDE solver) differentiable. Other key land-surface processes (e.g., those in Figure 1c) remain treated in their original, non-differentiable, piecewise form. As a result, the framework achieves gradient continuity for a subset of processes, but not necessarily full differentiability of the entire LSM.

The authors should clarify this scope explicitly in both the abstract and methods. Phrasing such as "a partially differentiable framework focusing on the heat–moisture solver" or "a differentiable core of Noah LSM" would be more accurate and prevent reader misinterpretation.

Response:

After careful consideration on the reviewer's concern, we respectfully maintain that the term "fully differentiable LSM" is appropriate in the context of modern automatic differentiation frameworks like PyTorch, upon which NoahPy is built. The reviewer correctly notes that many physical parameterizations in LSMs (e.g., for snow, vegetation, or surface roughness) are described by mathematically piecewise functions. In a strict sense, these are not continuously differentiable. However, our contribution is the re-implementation of the entire time-step solution of the modified Noah LSM—including its governing equations (as given in the process shown in Figure 1c) and all of its deterministic physical parameterizations—using native PyTorch operations.

In the context of deep learning frameworks, "differentiable" means that the automatic differentiation engine can compute a gradient for every operation in the computational graph. This is the same principle that allows for the training of deep neural networks using non-smooth activation functions like ReLU (Glorot et al, 2011). In Pytorch, non-differentiable points are handled using subgradient methods (https://docs.pytorch.ac.cn/docs/stable/notes/autograd.html), ensuring stable gradient propagation.

To make this important distinction clearer for all readers, we have added a clarifying statement to the Methods in Section 2.2. The new text reads:

"It is important to note that this includes all physical parameterizations, such as those for vegetation and snow processes shown in Figure 1c. While some of these processes are mathematically not differentiable, re-implementing them within PyTorch ensures that a valid gradient can be computed for every operation via the automatic differentiation engine. This makes the entire model fully differentiable in the context of gradient-based optimization." (lines 150-153)

**References:**

Paszke, A., Gross, S., Massa, F., Lerer, A., Bradbury, J., Chanan, G., Killeen, T., Lin, Z., Gimelshein, N., Antiga, L., Desmaison, A., Köpf, A., Yang, E., DeVito, Z., Raison, M., Tejani, A., Chilamkurthy, S., Steiner, B., Fang, L., Bai, J., and Chintala, S.: PyTorch: An Imperative Style, High-Performance Deep Learning Library, in: Proceedings of the 33rd International

Conference on Neural Information Processing Systems, Curran Associates Inc., Red Hook, NY, USA, 2019.

Glorot, X., Bordes, A., and Bengio, Y.: Deep Sparse Rectifier Neural Networks, In Proceedings of the fourteenth international conference on artificial intelligence and statistics, 315-323, 2011.

2. Gradient continuity within phase-dependent processes

The thermal conductivity λ, volumetric heat capacity $C_s$, and latent heat term Q exhibit abrupt transitions near the freezing point due to phase change. These discontinuities can interrupt or distort the backpropagated gradients, even if the overall framework is formally differentiable. The authors are encouraged to clarify how such non-smooth terms are handled in the current implementation.

Response:

We thank the reviewer for this insightful question. The reviewer is correct that the physics of phase change involves sharp discontinuities. Our implementation handles these non-smooth terms in a way that allows for continuous gradient propagation. To ensure this is clear to all readers, we have added a new paragraph to Section 2.2 (Implementation of NoahPy) that explicitly details how these phase-dependent processes are handled differentiably in our implementation:

"A specific example of this is the handling of phase-dependent processes. The Noah LSM handles the latent heat of fusion using a source term method, as represented by the Q term in the heat conduction equation (Equation 1). This term explicitly calculates and applies the latent heat required to be released or absorbed to keep the soil temperature at the freezing point during a phase change. While this represents an abrupt physical transition, numerically, this is not a true discontinuity but is implemented as a conditional logic check. In NoahPy, this entire conditional logic is re-implemented using a chain of native, computationally differentiable PyTorch operations, primarily torch. where, torch.min, and torch.max. PyTorch's automatic differentiation engine is designed to backpropagate through these subgradients, which is the same fundamental principle that enables the training of neural networks with

ReLU activations (Glorot et al.). This numerical implementation avoids a mathematical discontinuity. Therefore, PyTorch's autograd engine can compute a valid gradient through this logic." (lines 153-162)

**Reference:**

Glorot, X., Bordes, A., and Bengio, Y.: Deep Sparse Rectifier Neural Networks, In Proceedings of the fourteenth international conference on artificial intelligence and statistics, 315-323, 2011.

---

## Author Response (AR2)

**Response to Editor Mario Ebel**

Please ensure that the colour schemes used in your maps and charts allow readers with colour vision deficiencies to correctly interpret your findings. Please check your figures using the Coblis – Color Blindness Simulator (https://www.color-blindness.com/coblis-color-blindness-simulator/) and revise the colour schemes accordingly. --> Figs. 5, 6, 7

Reply: Thank you for your instruction. We changed the color schemes for figs 5, 6, 7.

[Figure]

Fig 5

[Figure]

Fig 6

Fig 7

Response to Anonymous referee #3

I agree with Reviewer 2's concern regarding the use of the term "fully differentiable Noah LSM". Only the soil temperature and moisture solvers are differentiated through this study. Based on my understanding, it is not very appropriate to call the model "fully differentiable" (or by nature, the Noah model is not a fully differentiable LSM). The current terminology would lead to some confusion. I would suggest changing the term in the title and throughout the text to be something like "A python-based Noah LSM with differentiable soil thermo-hydrologic solvers".

Reply: Thank you very much for your suggestion. We have modified the relevant text to reflect your concerns. Please see the change tracked doc for all modifications we have made. Note, the line numbering refers to that in the change tracked doc.

In Abstract (Ln 16-18):
To overcome this limitation, we present NoahPy, a  differentiable LSM developed by reconstructing the Noah LSM's governing partial differential equations into a process-encapsulated Recurrent Neural Network (RNN), with the heat–moisture solver forming the computational core.

Section 1 (Ln 70-72):
To address this gap, we introduce NoahPy: a  differentiable LSM built upon a version of the Noah LSM already modified and validated for simulating permafrost thermos-hydrology on the Qinghai-Tibet Plateau (QTP).

Section 2.2 (Ln 110-111)
The implementation of NoahPy involves recasting the numerical solution of the modified Noah LSM's governing equations into a  differentiable computational structure.
Ln 150-152
While some of these processes are mathematically not strictly differentiable, re-implementing them within PyTorch ensures that a valid gradient can be computed for every operation via the automatic differentiation engine. This makes the  model  differentiable in the context of gradient-based optimization.

Section 4 (Ln 370-371)
This study successfully demonstrated the development and application of NoahPy, a  differentiable land surface model for permafrost.

Section 5 (Ln 417-420)
In this study, we developed NoahPy, a  differentiable land surface model specifically improved for permafrost thermo-hydrology. We successfully recast the widely-used, Fortran-based Noah LSM into a "glass-box" Python framework that is both physically interpretable and implements differentiable operations  for gradient-based optimization.